# Structural Aspects and Intermolecular Energy for Some Short Testosterone Esters

**DOI:** 10.3390/ma15207245

**Published:** 2022-10-17

**Authors:** Alexandru Turza, Violeta Popescu, Liviu Mare, Gheorghe Borodi

**Affiliations:** 1Mass Spectrometry, Chromatography and Applied Physics Department, National Institute for R&D of Isotopic and Molecular Technologies, 67-103 Donat, 400293 Cluj-Napoca, Romania; 2Physics & Chemistry Department, Technical University of Cluj-Napoca, 28 Memorandumului Str., 400114 Cluj-Napoca, Romania; 3Molecular and Biomolecular Physics Department, National Institute for R&D of Isotopic and Molecular Technologies, 67-103 Donat, 400293 Cluj-Napoca, Romania

**Keywords:** 17β-hydroxyandrost-4-en-3-one, testosterone, ester, crystal structure, lattice energy, solubility

## Abstract

Testosterone (17β-hydroxyandrost-4-en-3-one) is the primary naturally occurring anabolic–androgenic steroid. The crystal structures of three short esterified forms of testosterone, including propionate, phenylpropionate, and isocaproate ester, were determined via single-crystal X-ray diffraction. Furthermore, all the samples were investigated using powder X-ray diffraction, and their structural features were described and evaluated in terms of crystal energies and Hirshfeld surfaces. They were also compared with the base form of testosterone (without ester) and the acetate ester. Moreover, from a pharmaceutical perspective, their solubility was evaluated and correlated with the length of the ester.

## 1. Introduction

Testosterone (17β-hydroxyandrost-4-en-3-one) is a cholesterol derivative and a naturally occurring anabolic steroid. It can be viewed as a derivative of the androstane group and the primary male sex hormone. It plays a major role in the development of male reproductive tissues and the maintenance of secondary male characteristics [1]. Testosterone has been shown to impact overall health and well-being [2] and prevent osteoporosis [3]. By binding to the androgen receptor, it exerts anabolic and androgenic properties that are the specific common characteristic of all derivatives belonging to this class [4]. In a medical context, testosterone is used to relieve symptoms of low testosterone in men (male hypogonadism) and breast cancer in women, as well as for hormone therapy in transgender men [5]. Testosterone targets androgen receptors [6], and previous studies have shown that normal testosterone levels in older men have an overall positive impact on health, decreasing body and visceral fat, increasing lean body mass, and improving cholesterol panel and carbohydrate metabolism [7]. Since it is an anabolic–androgenic steroid, testosterone is often used by athletes to increase performance [8]. Furthermore, medically, it can be used to relieve or treat protein degradation in certain catabolic states [9].

It is known that the testosterone base (Figure 1) has a short half-life of roughly a few hours; thus, it is often subjected to esterification in order to increase the half-life by intramuscular injections and avoid daily administration [10]. The esterified forms of testosterone possess a half-life ranging from around less than 1 day for testosterone acetate, 1 day for propionate, 2.5 days for phenylpropionate, and up to 3.1 days for testosterone isocaproate [11]. In this regard, the length of the ester can be correlated with the length of the carbon chain; thus, the longer the ester, the longer the half-life. 

The scheme of atoms and the labelling of steroid skeleton rings was made according to the established notations for the compounds belonging to this group [12] (Figure 1).

The current study aimed to investigate the characterisation of four testosterone prodrugs as follows:(i)Testosterone acetate (Androst-4-en-17β-ol-3-one 17β-acetate; TAce);(ii)Testosterone propionate (Androst-4-en-17β-ol-3-one 17β-propionate; TPro);(iii)Testosterone isocaproate (Androst-4-en-17β-ol-3-one 17β-4-methylpentanoate; TIso);(iv)Testosterone phenylpropionate (Androst-4-en-17β-ol-3-one 17β-phenylpropionate; TPhp).

For the last three, the crystal structures were determined and reported. 

The literature reports the crystal structure of a form of testosterone propionate that has a slightly different unit cell and does not have the positions of the hydrogen atoms reported [13], and another paper presents only the approximate parameters of the unit cell [14]. Other testosterone-based crystal structures that have been reported are testosterone buciclate [15], testosterone acetate [16], and testosterone base (the basic form without ester attached) [17]. This manuscript is aimed to investigate the structural features of these testosterone esters. 

A complete structural characterisation was carried out by means of X-ray single-crystal diffraction (for TPro, TPhp, TIso), X-ray powder diffraction, and the conformational analysis of steroid rings, and a quantitative measure of intermolecular interactions was accomplished by computation of lattice energies using the Coulomb–London–Pauli model corroborated with Hirshfeld surface analysis. Furthermore, computational methods were also applied to the testosterone base in order to be compared with the four esters.

As many pharmaceutical compounds, including various esterified forms of steroids, are labelled as poorly water-soluble but lipophilic agents, they might be dissolved in lipid-based preparations [18]. Based on this, the solubility in solutions of various organic oils was measured. The formulations of various drugs are currently available on the market, and all use oils as vehicles for certain compounds, including deoxycorticosterone, progesterone/oestradiol esters, testosterone esters with their analogues/derivatives, and vitamins such as K and E as well [19,20].

## 2. Materials and Methods

### 2.1. Materials and Crystallisation Experiments

Crystalline, white powders of esters for scientific research purposes, were received from Wuhan Shu Mai Technology Co., Wuhan, China and solvents from Merck, Taufkirchen, Germany.

All the investigated steroids were obtained at room temperature as white crystalline powders, which possess the possibility to be subjected to various recrystallisation methods. Suitable single crystals for X-ray data collection were successfully obtained in alcohols: methanol (TPro), ethanol (Tiso), and isopropyl alcohol (TPhp). 

Oils meeting the requirements of United States Pharmacopeia were received from Sigma-Aldrich (Taufkirchen, Germany), Tex Lab supply (USA), and Med Lab supply (USA).

### 2.2. X-ray Powder Diffraction (XRPD)

The samples were scanned on a Bruker D8 Advance diffractometer (Karlsruhe, Germany) having the X-ray tube set at 40 kV and 40 mA. The diffractometer is equipped with a germanium (1 1 1) monochromator used to obtain only the desired CuKα1 radiation and an LYNXEYE position-sensitive detector. X-ray diffraction patterns were recorded in the 3–40° (2θ) range using the DIFFRAC plus XRD Commander program, employing a scanning speed of 0.02°/s.

### 2.3. Single-Crystal X-ray Diffraction and Structural Refinement

The experimental single-crystal X-ray diffraction intensities were collected using a SuperNova diffractometer (Rigaku, Tokyo, Japan) equipped with dual microsources (Mo and Cu) and the X-ray tube operating at 50 kV and 0.8 mA. Data collection and reduction, Lorentz, polarisation, and absorption effect corrections were all performed with CrysAlis PRO software (Yarnton, Oxfordshire, UK) [21]. A multi-scan method using spherical harmonics in the SCALE3 abspack algorithm was applied for the empirical absorption correction. The crystal structures were solved as follows: Tiso and TPhp were solved with SHELXT [22] program using intrinsic phasing, and TPro was solved with direct methods by SHELXS [23]. Steroid structures were further refined via least squares minimisation with SHELXL [24] refinement package, and all programs were incorporated into Olex2 software (Durham, UK) [25]. 

Hydrogen atoms were geometrically located, treated, and refined as riding atoms, with the isotropic displacement parameter Uiso(H) = 1.2 Ueq(C) for ternary CH groups (C-H = 0.93 Å), secondary CH_2_ groups (C-H = 0.97 Å), and 1.5 Ueq(C) considered for all methyl CH_3_ groups (C-H = 0.96 Å).

### 2.4. Crystal Lattice Energy Computation and Hirshfeld and Fingerprint Plot Analyses

Based on the positions of atoms in the unit cell (determined by single-crystal X-ray diffraction technique), the classical atom–atom potential was calculated using the Coulomb–London–Pauli (CLP) model (Milan, Italy) [26]. The method evaluates crystal energies that can be divided into three distinct attraction terms, namely Coulombic, polarisation, and dispersion energies, and a fourth term that represents the repulsive component. 

Energy computation via the Coulomb–London–Pauli (CLP) approach involves pairs of individual atoms (*i*, *j*) that belong to different molecules and is the sum of four interaction terms according to Relation (1):(1)$$E_{ij}=1/(4\pi \varepsilon_{0})\left(q_{i}q_{j}\right)R_{ij}^{-1}-F_{P}P_{ij}R_{ij}^{-4}-F_{D}D_{ij}R_{ij}^{-6}+F_{R}T_{ij}R_{ij}^{-12}$$
(2)$$\;q_{i}=F_{Q}q_{i}^{0}$$

The Coulombic energy is the first term, polarisation energy is the second term, the dispersion term is the third one, and the last term is repulsion.

The *F_Q_*, $\;F_{P}$,$\;F_{D}$, and $F_{R}$ coefficients involved in Relations (1) and (2) are empirically disposable scaling parameters and the $P_{ij}$, $D_{ij}$, and $T_{ij}$ coefficients depend on the local vicinity of the atom in the molecule.

The Coulombic component is treated according to Coulomb’s law, the polarisation term is estimated in the approximation of the linear dipole, the dispersion energy is approximated as the inverse of the distance at the sixth power, and the repulsive term is due to the modulation of the overlapping wave function.

Molecular 3D Hirshfeld surfaces and their related 2D fingerprint plots were generated by CrystalExplorer software (Perth, Australia) [27] based on the *d_norm_* function, which can be expressed in Relation (3).
(3)$$d_{norm=}\frac{d_{i}-r_{i}^{vdW}}{r_{i}^{vdW}}+\frac{d_{e}-r_{e}^{vdW}}{r_{e}^{vdW}}$$
where *d_e_* is the distance from the surface to the nearest external nucleus, while *d_i_* represents the distance from the surface to the nearest nucleus inside the surface. The fingerprint plots are a 2D diagram, where *d_i_* and *d_e_* are represented in order to identify the nature and types of different intermolecular contacts [28].

### 2.5. Solubility Check

The solubility for the four esters (mg/mL) was measured in solutions of various organic oils: medium-chain triglyceride (MCT), grape seed oil (GSO), castor oil, cottonseed oil, apricot oil, and sesame oil. 

Each solution was composed of a mixture of benzyl benzoate, benzyl alcohol, and oil having a volumetric ratio of 78% oil, 20% benzyl benzoate, and 2% benzyl alcohol. In various pharmaceutical preparations of lipophilic compounds, including various steroids, benzyl benzoate is used as a solubiliser (co-solvent), benzyl alcohol acts as a solvent and at the same time prevents microbial growth and increases the lipid solubility of various esterified compounds, while the oils are used as carriers.

The solubility evaluation was performed in multiple steps at room temperature (25 °C) by successively adding small amounts of raw materials (2–5 mg each step), and the solution was stirred for up to several hours until dissolved. When it was found that excess raw material remained (in suspension), small amounts of solution (mixture of benzyl benzoate, benzyl alcohol, and oil) were added until the resulting solution became perfectly transparent and clear. In order to obtain good accuracy, three such procedures were carried out, and their average was used.

## 3. Results and Discussion

### 3.1. Crystal Structures and Supramolecular Descriptions

The good agreement generated between the experimental powder X-ray diffraction patterns and the simulated patterns based on the positions of the atoms in the unit cell shows a good structural homogeneity and that the studied single crystals are representative of the entire bulk of samples (see Appendix A). 

The details with regard to single-crystal data and refinement for the studied esters are given in Table 1.

The aim was to determine the absolute configurations for each of the testosterone esters investigated. The values of Flack parameters of 004(6) for TPro and 0.06(8) for TIso confirm the correctness of the absolute configurations; on the other hand, for TPhp, the negative value of the Flack parameter shows that this parameter has no meaning. 

#### 3.1.1. TAce (Testosterone Acetate)

The acetate ester is the shortest esterified steroid ester available and is also the shortest testosterone ester available. The CSD database contains one entry reporting only the cell parameters for this particular testosterone ester [16] and one entry reporting the unit cell parameters and atomic coordinates [14].

The acetate ester was found to crystallise in the noncentrosymmetric orthorhombic P2_1_2_1_2_1_ space group with one molecule in the asymmetric unit (Appendix A) and four in the unit cell. The carbonyl O1 oxygen of the ketone group is involved in C-H•••O bifurcated hydrogen bonds and contributes to crystal stability. One is made with a neighbouring five-membered ring (C15-H15A•••O3) and one towards the CH_3_ of terminal methyl in the acetate group (Appendix A). Hydrogen bonding distances are presented in Appendix A.

#### 3.1.2. TPro (Testosterone Propionate)

Compared with testosterone acetate, the propionate ester is characterised by an ester chain with an extra carbon atom. The crystal structure of propionate ester was previously reported [13] but has slightly smaller unit cell parameters and lacks hydrogen atoms. Similar to TAce, it crystallises in the orthorhombic P2_1_2_1_2_1_ space group, with one molecule in the asymmetric unit (Figure 2a), the unit cell hosting four such molecules. Considering the H atoms located in idealised positions via X-ray crystallography, apparently, it seems that only the C6-H6B**•••**O1 interaction between the O1 ketone group and the neighbouring D ring in the structure has a separating distance shorter than the sum of van der Waals radii and plays a role in stability. In reality, having the H atoms normalised (the C-H distance is 1.089 Å), the ketone O1 oxygen is involved in bifurcated C-H**•••**O interactions, with the second being C16-H16A**•••**O1 with d(H**•••**O) = 2.6185 Å. This distance is close to the sum of the van der Waals radii with 1.20 Å for hydrogen and 1.52 Å for oxygen [29]. An overall packing perspective is shown in Figure 2b. 

#### 3.1.3. TIso (Testosterone Isocaproate)

The asymmetric unit (Figure 3a) consists of only one steroid molecule and was found to crystallise in the noncentrosymmetric P2_1_ monoclinic space group. Ketone O1 participates in the formation of supramolecular self-assemblies, being involved in the trifurcated C-H**•••**O hydrogen bonding. One interaction is formed between a neighbouring six-membered B ring (C6-H6B**•••**O1), one binds the O1 carbonyl oxygen with a five-membered ring (C16-H16B**•••**O1), while the last bridges the methyl CH_3_ group (C19-H19A**•••**O1); all these interactions are detailed in Appendix A. An overall packing diagram shows the self-arrangements of steroid molecules in layers (Figure 3b).

#### 3.1.4. TPhp (Testosterone Phenylpropionate)

It was found that the steroid crystallises in the monoclinic P2_1_ space group with one molecule in the asymmetric unit (Figure 4a) and two in the unit cell. Similar to testosterone propionate, considering the H positions determined via X-ray diffraction, it seems only one C-H**•••**O interaction exists in the crystal lattice (C24-H24**•••**O1 between the terminal phenyl ring and the ketone O1 oxygen, Appendix A) that is shorter than the sum of van der Waals radii. After the normalisation of the C-H distances, there is C2-H2A**•••**O3 interaction between ring A and carbonyl O3 oxygen, which has d(H**•••**O) = 2.714 Å, situated just at the limit distance of 2.72 Å [29]. The overall packing perspective of testosterone phenylpropionate is presented along the ob-axis (Figure 4b). 

From the analysis of the crystal structures, the following conclusions can be summarised:(i)Asymmetric units are characterised by a single molecule for each ester;(ii)The formations of supramolecular 3D assemblies are to some extent driven by the C-H**•••**O interactions, although the dispersion energy has the greatest weight, as will be shown in the crystal energies analysis section; donor–acceptor separation distances show similar values to those of other crystals driven by C-H**•••**O interactions and belong to the steroid family [30,31,32,33,34];(iii)The six-membered A rings are found in the intermediate sofa-half-chair geometry, and the B and C rings show chair conformations, while the five-membered D rings adopt intermediate envelope-half-chair geometry. Similar geometries of skeleton rings have been reported in the crystal structure of its C-17 methylated form [35].

In Figure 5, the overlap of the molecular structures is exemplified. For example, (TPro, Tphp), which differ only by the extra phenyl ring of TPhp, show a totally different orientation of the tails, as in the case of the (TAce, TIso) pair as well. It can be seen that the part of the molecules representing testosterone, the base of the ester structures, overlaps very well in all pairs. Instead, there are differences in the orientation of carbon tails. Thus, for the pair (TPro, TPhp), the C17-O2-C20-O3 torsion angle is 2.89° for TPro and −3.78° for TPhp. The angle between the planes defined by the O3-C20-O2 atoms for the pair (TPro, TPhp) is 64.93°. For the pair (TAce, TIso), the C17-O2-C20-O3 torsion angle is 3.73° for TAce and 4.93° for TIso, and the angle between the planes defined by the O3-C20-O2 atoms for the two structures is 75.03°. 

The overlap of (TAce, TPhp) and (TPro, TIso) pairs show a better match; thus, the angle between the O3-C20-O2 planes is 25.35° for (TAce, TPhp) and 11.22° for (TPro, TIso).

### 3.2. Crystal Energy Analysis 

The total crystal lattice energies, as well as the nature and magnitudes of the individual four energy components, were computed using the atom–atom Coulomb–London–Pauli model, and the results are shown in Table 2. Moreover, the energies of the four testosterone esters were compared with those of the testosterone base [17] deposited in CSD (denoted Tbas), which is not esterified. 

All four steroid structures are characterised by large values of dispersion energies, and this component is dominant. As a general trend, the dispersion energies were found to be more significant, as the ester chain is longer; thus, TAce, which is the shorter ester, has a value of −126.1 kJ/mol, whereas TPhp, which represents the longest ester, is −149.0 kJ/mol. 

Due to the fact that the structures lack strong hydrogen bonds, the Coulombic energy contributes the least in the crystal packings, with similar values through all the crystals, which are in the range of −15.5 kJ/mol for TAce and −21.3 kJ/mol for TIso. Carbonyl**•••**hydrogen interactions were found in the Coulombic term and for the derivatives under study, the Coulombic energy has a weight between 9.7% (for TAce) and 12.6% (for TIso).

By contrast, the Coulombic component (−33.3 kJ/mol) in the testosterone base (without ester), which presents strong classical O-H**•••**O hydrogen bonds, contributes more to the lattice energy.

Polarisation and repulsive components do not display a particular trend, but on the other hand, the total lattice energy becomes lower as the steroid molecular mass increases. The shortest ester (TAce) has a total lattice energy of −159.5 kJ/mol and TPhp, which is the longest ester, has an energy of −184.5kJ/mol and the most bound structure of all five steroids. 

Similar crystal lattice behaviour in the sense that the dispersion term dominates the crystals, and the total energy becomes greater with the increase in ester length has been previously reported in other anabolic–androgenic agents from the steroid group [36,37,38,39].

### 3.3. Hirshfeld and Fingerprint Plot Analysis 

The molecular 3D Hirshfeld surfaces of the studied esters (Appendix A) were generated based on d_norm_ and were compared with those of the base form (TBas) (Appendix A). As the asymmetric unit of TBas is characterised by two individual molecules, they were analysed separately. 

The surfaces are interactively illustrated with arrows for the intermolecular C-H**•••**O and O-H**•••**O contacts with distances shorter than the sum of the van der Waals radii, which are listed in Appendix A. 

The surfaces can be understood by colour code; specifically, the red areas illustrate the intermolecular contacts with distances smaller than the sum of the van der Waals radii, the white indicates separation distances approximately equal to vdW radii, and the blue areas show the contacts with longer distances. 

For each crystal, a 2D fingerprint plot (Figure 6) is generated, which is a transposition of the 3D Hirshfeld surface. The fingerprint plot of the base form (TBas) shows two fingerprint diagrams.

The analyses of 3D Hirshfeld surfaces, their related 2D fingerprint plots shapes, and (d_e_ and d_i_) distances summarise the following structural features: (i)Fingerprint plots of esterified forms (Figure 6b) (TAce, TPro, TPhp, and TIso) show symmetry in the spikes, which is a particular feature for the crystals with one molecule in asymmetric units, while the plots of TBas (Figure 6a) are asymmetric due to the different molecular environment in the crystal;(ii)The diagrams of Tace, TPro, and TIso illustrate protruding H**•••**O/O**•••**H spikes, denoting the presence of C-H**•••**O hydrogen bonds, while for TPhp, the lack of H**•••**O/O**•••**H spikes shows that the separation distances of the C-H**•••**O interactions fall in a range closer to the sum of vdW radii;(iii)The fingerprint plots of Tbas show more protruding H**•••**O/O**•••**H spikes compared with its esterified forms and suggest that strong O-H**•••**O interactions play more important roles in packing; this feature is seen in the evaluation of crystal energies where the Coulombic energy becomes more significant in TBas due to the presence of strong O-H**•••**O contacts;(iv)The quantitative breakdown of fingerprint diagrams (Table 3) in all five crystals reveals a high percentage of H**•••**H contacts, medium contribution by O**•••**H/H**•••**O intercontacts and considerably smaller for C**•••**H/H**•••**C, respectively;(v)The large percentages in H**•••**H contacts for all five structures (fingerprint plots breakdown in Table 3) corroborated with the crystal energies (Table 2) are suggesting that dispersion effects play the major role.

### 3.4. Solubility Check 

The values obtained by solubility evaluation are summarised in Appendix A and are graphically represented in Figure 7. 

Depending on the length of the ester, it is observed that the shortest ester (the acetate) has the lowest solubility, while the longest ester (isocaproate) has a roughly four-fold greater solubility. Propionate and phenylpropionate esters have similar and slightly higher values than acetate. Although phenylpropionate has six more carbon atoms than propionate, the solubilities are similar, so it can be noted that what matters in increasing solubility is the length of the chain (the phenyl ring does not lead to an increase in solubility).

It is worth mentioning that solubility correlates with the half-life of the prodrug, so acetate, which has the shortest half-life, has the lowest solubility, and isocaproate, which has the longest half-life, has the highest solubility.

Out of the six mixtures analysed, it is observed that castor oil can support the highest solubility without crashing (crystallisation of compound) and to a lesser extent MCT (for TAce, TPro, and TPhp), at the same time behaving as solvents as well. This is interesting because MCT is characterised by the lowest value of the viscosity coefficient, while castor oil is the most viscous.

The literature reports the solubility of hydroxyprogesterone caproate polymorphs in castor oil at a temperature of 20 °C, which are 278 mg/mL and 301 mg/mL [40]. Another method used for the development of the preparations of drugs with poor water solubility (including various esterified forms of steroids) is by oil-in-water (*o/w*) microemulsions. For example, a previous study shows that a microemulsion based on soybean oil and dimethoxytetraethylene glycol supports concentrations of 3.42 mg/mL (for testosterone propionate), 31.5 mg/mL (for testosterone enanthate), and 2.16 mg/mL (for medroxyprogesterone acetate) in soybean oil. In dimethoxytetraethylene glycol, higher concentrations of 12 mg/mL (for testosterone propionate) 91.2 mg/mL (for testosterone enanthate) and 1.32 mg/mL (for medroxyprogesterone acetate) were obtained [41].

## 4. Conclusions

The crystal structures of three testosterone-based esters (propionate, phenylpropionate, and isocaproate) were determined; they belong to the noncentrosymmetric monoclinic P2_1_ and orthorhombic P2_1_2_1_2_1_ space groups. 

These three esters were further analysed and compared with the acetate ester and with the nonesterified base form. Backbone steroid rings possess similar geometry in all compounds, the A steroid rings adopt intermediate sofa-half-chair conformations, and the B and C rings have chair conformations, while ring D depicts an intermediate envelope-half-chair conformation. By overlapping the esters, a very good match of the steroid skeleton rings (the base of the ester structures) emerges, and the major structural differences are manifested in the orientation of the tails. 

Computational methods showed that in all crystal structures, the supramolecular arrangements and crystal stability are characterised and assured by dominant dispersion effects, and the total lattice energies are greater in absolute terms, as the ester chain is longer, while C-H•••O hydrogen bonds in all esters play a less important role. 

The solubility of the four derivatives was tested to evaluate the changes based on the added ester functionalities, and it was found that the shortest ester (acetate) has the lowest solubility, while the longest ester (isocaproate) is roughly four times greater; meanwhile, propionate and phenylpropionate are between the two and show similar values. 

## Figures and Tables

**Figure 1 materials-15-07245-f001:**
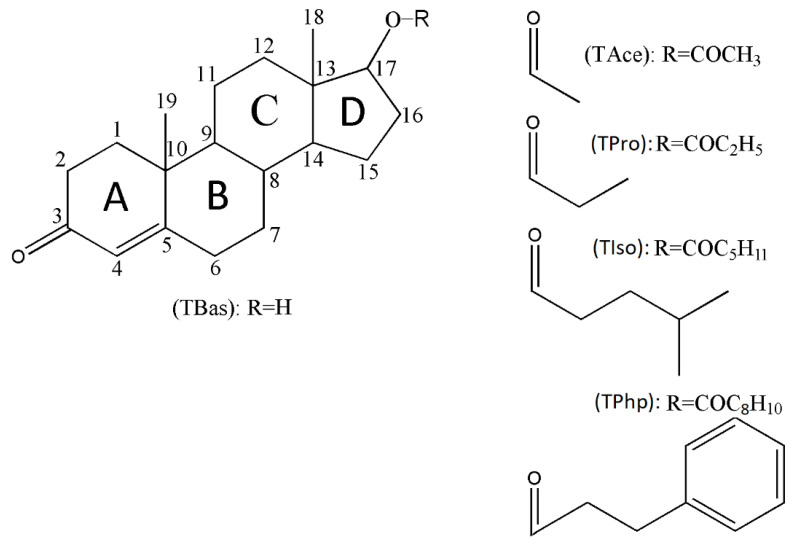
Chemical structures of 17β-hydroxyandrost-4-en-3-one (testosterone) displaying the steroid backbone labelling system and other studied testosterone-based steroids.

**Figure 2 materials-15-07245-f002:**
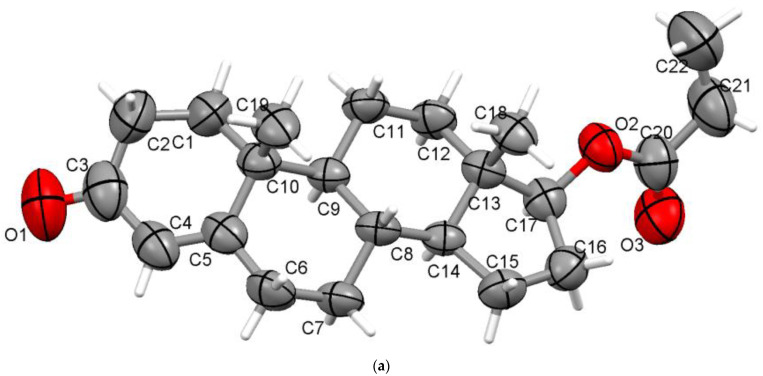
Asymmetric unit of TPro presenting nonhydrogen atoms at 50% probability level (**a**) and overall packing diagram along a-axis (**b**).

**Figure 3 materials-15-07245-f003:**
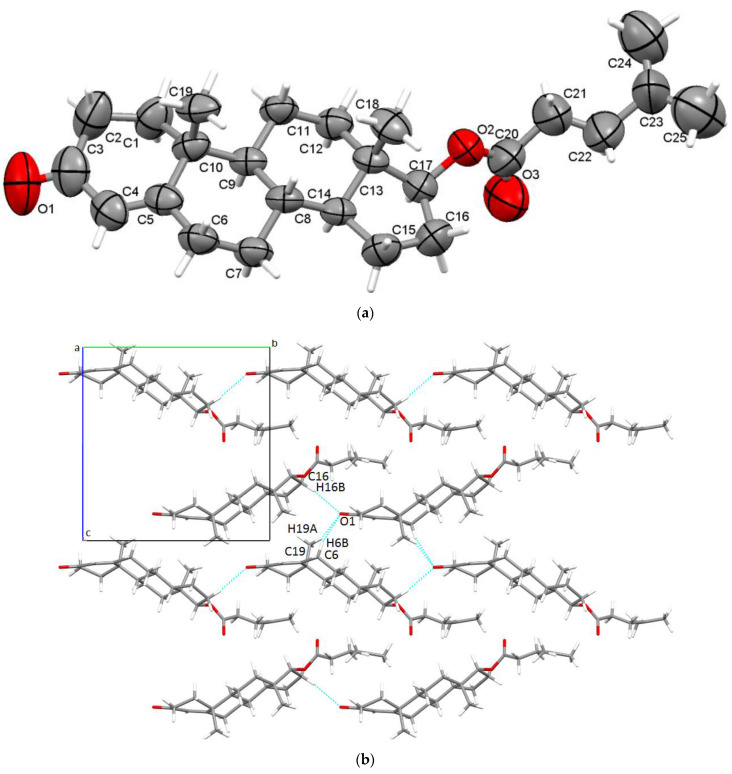
Asymmetric unit of TIso presenting nonhydrogen atoms at 50% probability level (**a**) and overall packing diagram along a-axis (**b**).

**Figure 4 materials-15-07245-f004:**
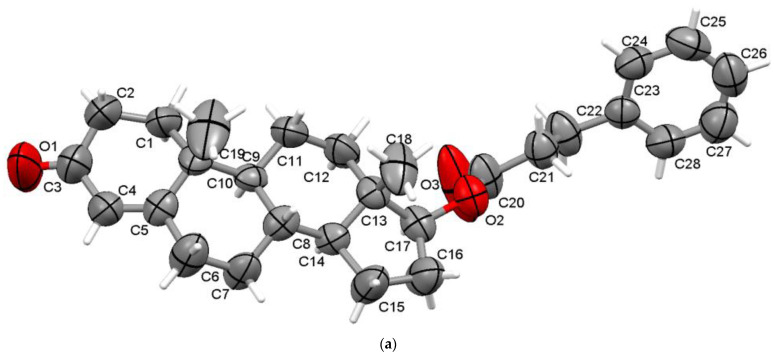
Asymmetric unit of TPhp presenting nonhydrogen atoms at 50% probability level (**a**) and overall packing diagram along b-axis (**b**).

**Figure 5 materials-15-07245-f005:**
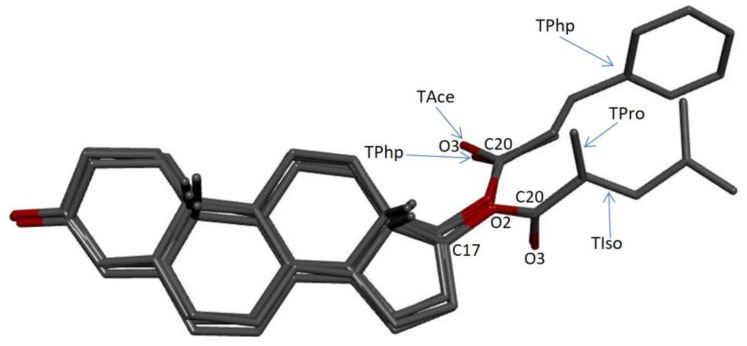
Molecular overlaps of the investigated esters.

**Figure 6 materials-15-07245-f006:**
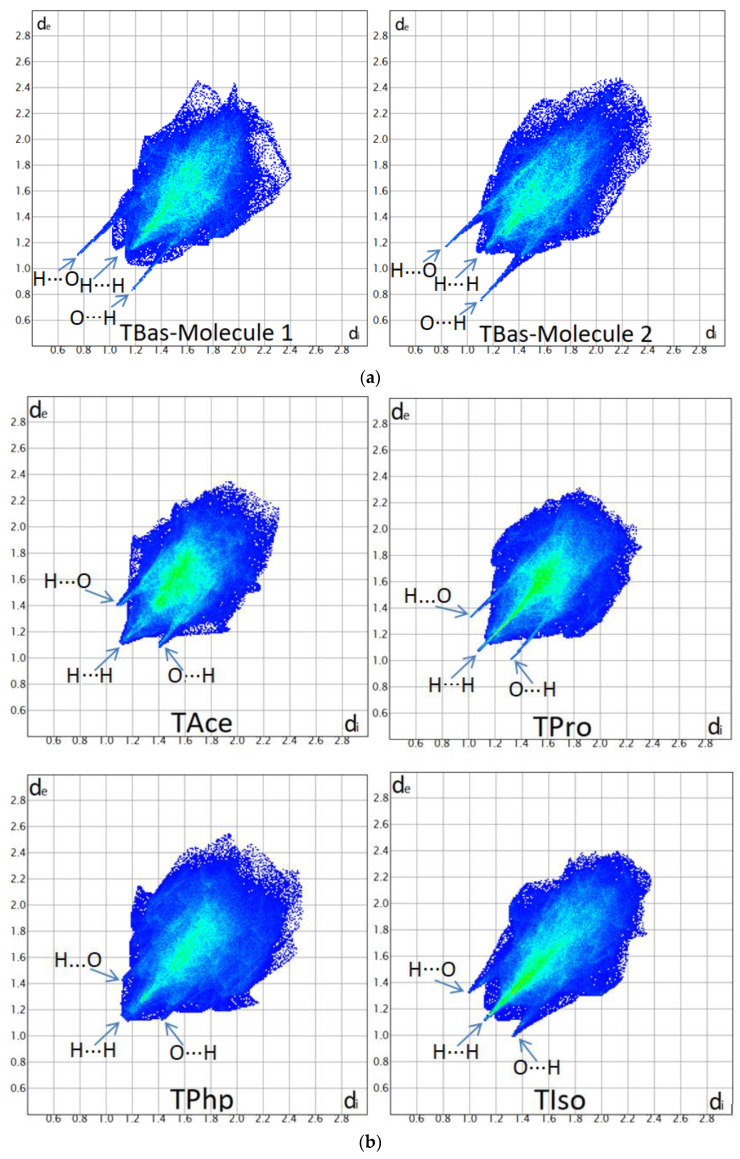
Fingerprint plots displaying close contacts in studied crystals: TBas (**a**) and studied esters (**b**).

**Figure 7 materials-15-07245-f007:**
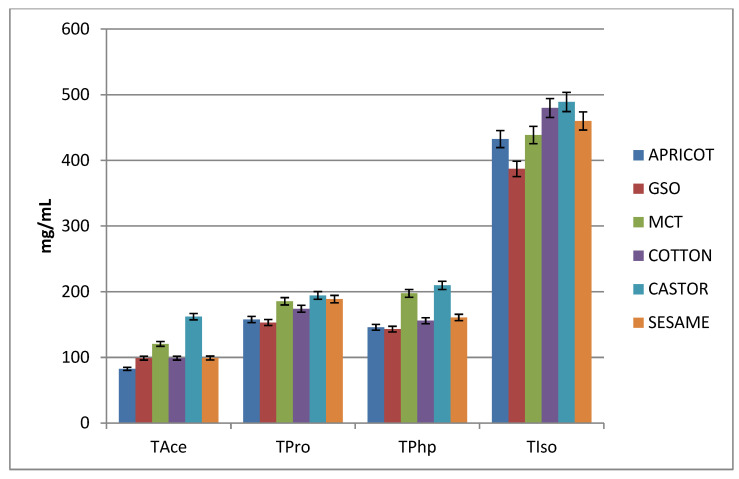
Graphical representations of ester solubility.

**Table 1 materials-15-07245-t001:** Crystal structures and refinement data of investigated esters.

Identification Code	TPro (Testosterone Propionate)	TIso (Testosterone Isocaproate)	TPhp (Testosterone Phenylpropionate)
Empirical formula	C_22_H_32_O_3_	C_25_H_38_O_3_	C_28_H_36_O_3_
Formula weight	344.47	386.57	420.57
Temperature/K	293(2)	293(2)	293(2)
Crystal system	orthorhombic	monoclinic	monoclinic
Space group	P2_1_2_1_2_1_	P2_1_	P2_1_
a/Å	7.57470(16)	7.2877(3)	13.4097(7)
b/Å	12.6768(2)	12.3741(5)	5.9105(3)
c/Å	20.4038(4)	13.1272(6)	15.4054(8)
α/°	90	90	90
β/°	90	103.305(4)	95.073(5)
γ/°	90	90	90
Volume/Å^3^	1959.23(6)	1152.02(9)	1216.22(11)
Z	4	2	2
ρcalcg/cm^3^	1.168	1.106	1.148
μ/mm^−1^	0.594	0.552	0.568
F(000)	752.0	418.0	456.0
Crystal size/mm	0.09 × 0.08 × 0.07	0.1 × 0.03 × 0.01	0.09 × 0.09 × 0.01
Radiation	CuKα (λ = 1.54184)	CuKα (λ = 1.54184)	CuKα (λ = 1.54184)
2Θ range/°	8.212 to 141.254	9.952 to 141.104	5.76 to 141.522
Index ranges	−8 ≤ h ≤ 9, −15 ≤ k ≤ 15, −24 ≤ l ≤ 24	−8 ≤ h ≤ 8, −15 ≤ k ≤ 15, −15 ≤ l ≤ 16	−16 ≤ h ≤ 16, −7 ≤ k ≤ 7, −18 ≤ l ≤ 18
Reflections collected	28,102	15,810	14,176
Independent reflections	3723 [R_int_ = 0.0243, R_sigma_ = 0.0119]	4322 [R_int_ = 0.0246, R_sigma_ = 0.0185]	4529 [R_int_ = 0.0796, R_sigma_ = 0.0508]
Data/restraints/ parameters	3723/0/229	4322/1/257	4529/1/282
Goodness-of-fit on F2	1.044	1.076	1.078
Final R indexes [I ≥ 2σ (I)]	R_1_ = 0.0421, wR_2_ = 0.1187	R_1_ = 0.0621, wR_2_ = 0.1739	R_1_ = 0.0762, wR_2_ = 0.1784
Final R indexes [all data]	R_1_ = 0.0440, wR_2_ = 0.1217	R_1_ = 0.0740, wR_2_ = 0.1900	R_1_ = 0.1102, wR_2_ = 0.1959
Largest diff. peak/hole/e Å^−3^	0.17/−0.17	0.33/−0.21	0.25/−0.21
Flack parameter	0.04(6)	0.06(8)	−0.4(3)

**Table 2 materials-15-07245-t002:** Crystal energies of base form and its four esters.

Structure	Molar Mass g/mol	E_coul_ (kJ/mol)	E_pol_ (kJ/mol)	E_disp_ (kJ/mol)	E_rep_ (kJ/mol)	E_latt_ (kJ/mol)
TBas	288.43	−33.3	−47.5	−130.9	60.9	−150.8
TAce	330.46	−15.5	−55.0	−126.1	37.1	−159.5
TPro	344.49	−18.3	−55.5	−126.4	33.9	−166.3
TIso	386.57	−21.3	−57.7	−141.9	52.7	−168.2
TPhp	420.59	−19.3	−52.3	−149.0	36.1	−184.5

E_coul_: the Coulombic term; E_pol_: the polarisation term; E_disp_: the dispersion term; E_rep_: the repulsive term; E_latt_: total crystal lattice energy.

**Table 3 materials-15-07245-t003:** Contributions to the Hirshfeld surfaces for various intercontacts.

Structure	H•••H	O•••H/H•••O	C•••H/H•••C	C•••O/O•••C	O•••O	C•••C
TBas Mol A	78.8%	17.2%	4.0%	-	-	-
TBas Mol B	76.4%	19.7%	3.9%	-	-	-
TAce	75.4%	20.2%	2.8%	1.2%	-	0.4%
TPro	76.5%	19.4%	4.0%	-	0.1%	-
TIso	80.4%	16.3%	3.3%	-	-	-
TPhp	72.7%	15.2%	12.1%	-	-	-

## Data Availability

Data is contained within the article or Appendix A.

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
