# Peer review of "Structural Aspects and Intermolecular Energy for Some Short Testosterone Esters"

_materials, 2022, doi:10.3390/ma15207245_

Round 1

Reviewer 1 Report

In this manuscript, the authors reported their latest research article titled ‘Structural aspects and intermolecular energy for some short testosterone esters’ in the Materials. They determined crystal structures of three esterified forms of testosterone including propionate, phenylpropionate and isocaproate ester by single crystal X-ray diffraction. They investigated all samples by powder X-ray diffraction and structural features and evaluated in terms of crystal energies and Hirshfeld surfaces. They also were compared with the base form of testosterone (without ester) and the acetate ester. Overall, this manuscript provides interesting insight along with comprehensive examinations of the material parameters. On the other hand, there are some comments that need to be addressed before the publication.

1.    Further comments can be made about the reason why TPhp's Flack parameter value is negative. It should also be further explained why absolute configuration is not obtained for TPhp and why a meaningless Flack parameter value is obtained.

2.    Calculations can be expressed in more detail.

3.    Why are dispersion energies more important because the ester chain is longer?

4.    It has been said that MCT and castor oil support the highest resolution, but this does not apply to TIso, so the sentence should be expressed differently.

5.    The current version of the manuscript lacks detailed literature and needs to compare the presented material with the other state-of-the-art material systems for drug-based applications. A comparison table would help more for this explanation in the conclusion part.

Author Response

The authors would like to thank to the reviewer for the time and valuable guidance, which contributes to the improvement of the manuscript.

According to the instructions, the corresponding modifications and additions were made. Changes and additions are highlighted in text.

In this manuscript, the authors reported their latest research article titled ‘Structural aspects and intermolecular energy for some short testosterone esters’ in the Materials. They determined crystal structures of three esterified forms of testosterone including propionate, phenylpropionate and isocaproate ester by single crystal X-ray diffraction. They investigated all samples by powder X-ray diffraction and structural features and evaluated in terms of crystal energies and Hirshfeld surfaces. They also were compared with the base form of testosterone (without ester) and the acetate ester. Overall, this manuscript provides interesting insight along with comprehensive examinations of the material parameters. On the other hand, there are some comments that need to be addressed before the publication.

  1. Further comments can be made about the reason why TPhp's Flack parameter value is negative. It should also be further explained why absolute configuration is not obtained for TPhp and why a meaningless Flack parameter value is obtained.

Answer: The determination by anomalous dispersion is effective in compounds which contain lighter and heavier atoms as well. The negative value has no meaning and most likely appears because the errors.  Since it is difficult to determine the absolute configuration for compounds containing light elements (C, O, N) it is likely that TPhp is related to some problems regarding this particular aspect.

  1. Calculations can be expressed in more detail.

Answer: New details regarding the computational part have been added.

  1. Why are dispersion energies more important because the ester chain is longer?

Answer: The dispersion energy increases as the number of contact points between molecules increases and for larger molecules it seems that the number of interactions is greater. Although dispersion forces between two individual molecules is weak and decreases steeply with distance (R), in crystals the effects are felt cumulatively. The dispersion term appears due to the induced dipoles and if we have a larger molecule, probably the neighboring atoms interact and create more induced dipoles.

  1. It has been said that MCT and castor oil support the highest resolution, but this does not apply to TIso, so the sentence should be expressed differently.

Answer: The paragraph was corrected as follows: “Out of the six mixtures analyzed it is observed that castor oil can support the highest solubility without crashing (crystallization of compound) and to a lesser extent MCT (for TAce, TPro, TPhp), at the same time behaving as solvents as well."

  1. The current version of the manuscript lacks detailed literature and needs to compare the presented material with the other state-of-the-art material systems for drug-based applications. A comparison table would help more for this explanation in the conclusion part.

Answer: The literature in general is poor regarding the study of the solubility of different lipophilic compounds/prodrugs/preparations. However, the solubility of the compound hydroxyprogesterone caproate at a temperature of 20C was reported and this was added to the text. Another method used for development of preparations of drugs with poor water solubility (including various esterified forms of steroids) is by oil-in-water (o/w) microemulsions. For example, a previous study shows that such a microemulsion based on soybean oil and dimethoxytetraethylene glycol supports concentrations of 3.42 mg/mL (for testosterone propionate), 31.5 mg/mL (for testosterone enanthate) and 2.16 mg/mL (for medroxyprogesterone acetate) in soybeam oil . In dimethoxytetraethylene glycol were obtained higher concentrations of 12 mg/mL (for testosterone propionate) 91.2 mg/mL (for testosterone enanthate) and 1.32 mg/mL (for medroxyprogesterone acetate)

Reviewer 2 Report

Article “Structural aspects and intermolecular energy for some short testosterone esters” by Turza and coauthors presents crystal structures, their analysis and characterization of solubility for three testosterone esters.

Firstly, the article as such is a typical crystallographic paper, which, in case of studying biologically important compounds, often presents also some physical characterization, particularly solubility determination. Therefore, the authors choice to submit it to journal Materials, particularly to special issue “Synthesis, Characterization and Applications of Sustainable Advanced Nanomaterials” is not clear, as the article does not fit in this topic or in this journal. Therefore, I believe transferring the article to journal Crystals is necessary to fit in the correct field and for article to reach appropriate audience.

Secondly, it seems that the authors are not well familiar with the way how such crystallographic articles are often prepared for similar level crystallography journals in the recent years. Therefore, I would recommend changes to the article to make it more appropriate for publication and concise:
a. In labels of Figures 3 – 6 use along a-axis and along b-axis.
b. Regarding conclusion (i) in P9 – if pure enantiomers were crystallized, this is he only viable option and would be impossible otherwise.
c. Regarding conclusion (iv), I believe instead of configuration the authors mean conformation (as configuration is as taken, while conformation is the spatial arrangement related to potential to take specific geometry in the crystal). Also use “do not have structure voids”. I also do not see anything awkward related to conformation, it is just specific to steroids.
d. Figure 7 presents conformation and directly copes Fig. 3 – Fig. 6. Instead, if similarity in conformation is the thing which is intended to be shown, make actual overlay of conformations so the identical parts are actually above one another. This can be done in, e.g., Discovery Studio and can be tried to achieved in Mercury as well.
e. Section 3.2. should be added as just one line to beginning of Section 3.1 indicating the agreement of XRPD (characterizing bulk sample) and from structure simulated XRPD (characterizing the analyzed crystal) showing that crystals represent the bulk sample. Figure 8 should be transferred to Supporting Information. The would deserve a place in the article only in case there would be complete mismatch between the patterns that would deserve some detailed explanation.
f. not directly related to crystallography but more to organic chemistry articles: Figures 1 and 2 would be better to join in one Structure presented in one figure by showing testosterone backbone with atom numbering and R in place of hydroxyl group attached to C17, and below the structure giving that R = OH in case of testosterone, COCH3 (could better be using structural formula) in case of acetate etc. Also, label chemical structures as “chemical structure” instead of “Structural/chemical perspective”.
g. If the article would be published in crystallography journal, explanation s regarding Hirshfeld surface could be reduced (e.g, reason why there are 2 surfaces for TBas, L324-327).
h. Figure 10 could be transferred to Supporting Information, as it is not discussed in the article text, but more easily interpretable 2D Fingerprint plot is the one being actually analyzed. Also the Table 3 showing information on mostly weak hydrogen bonds is not necessary in the text and I recommend to transfer it to Supporting Information.

Minor comments:

1. Introduction related to physiological action seems too long for such a crystallography paper.

2. L51-53 in P2 does not seem necessary in a scientific article.

3. “USP grade oils (they meet the requirements of United States Pharmacopeia)” repeats the same information twice, use either only “USP grade olis” or “Oils meeting the requirements of United States Pharmacopeia”

4. LynxEye is a position sensitive detector (L110)

5. As obtained from three measurements, add error bars (standard deviation or uncertainty) to Figure 11. I also strong;y encourage authors not to repeat information in Table 5 and Figure 11 and to transfer the former one to the Supporting Information.

6. Provided it was one of the aims, give the absolute conformation for TPro and Tiso determined. Based on the conclusion “The Flack parameter shows that the propionate and acetate esters are pure enantiomers.” It seems that it possible that the third compound could not be pure enantiomer. Is it likely so or is it more plausible that this is just a result of some problems with crystal structure determination regarding this particular aspect?

7. As the structure of Testosterone acetate was not determined in this work, I suggest that Figure 3 can be put in Supporting Information. The most important structure data and its analysis can still be presented in the paper.

8. There is a typo in Table 2 for value 33.3 of testosterone.

9. In L384 if you write “On the other hand” you should have “On the one hand” somewhere before.

Author Response

The authors would like to thank to the reviewer for the time and valuable guidance, which contributes to the improvement of the manuscript.

According to the instructions, the corresponding modifications and additions were made. Changes and additions are highlighted in text.

Article “Structural aspects and intermolecular energy for some short testosterone esters” by Turza and coauthors presents crystal structures, their analysis and characterization of solubility for three testosterone esters.

Firstly, the article as such is a typical crystallographic paper, which, in case of studying biologically important compounds, often presents also some physical characterization, particularly solubility determination. Therefore, the authors choice to submit it to journal Materials, particularly to special issue “Synthesis, Characterization and Applications of Sustainable Advanced Nanomaterials” is not clear, as the article does not fit in this topic or in this journal. Therefore, I believe transferring the article to journal Crystals is necessary to fit in the correct field and for article to reach appropriate audience.

Secondly, it seems that the authors are not well familiar with the way how such crystallographic articles are often prepared for similar level crystallography journals in the recent years. Therefore, I would recommend changes to the article to make it more appropriate for publication and concise:
a. In labels of Figures 3 – 6 use along a-axis and along b-axis.

Answer: this was corrected
b. Regarding conclusion (i) in P9 – if pure enantiomers were crystallized, this is he only viable option and would be impossible otherwise.

Answer: This indication is true and the overwhelming majority of compounds in the steroid class (in CSD) are non-centrosymmetric. However, there are few reported structures that are centrosymmetric. That is why we mentioned point (i) as a conclusion.

Point (i) was removed.
c. Regarding conclusion (iv), I believe instead of configuration the authors mean conformation (as configuration is as taken, while conformation is the spatial arrangement related to potential to take specific geometry in the crystal). Also use “do not have structure voids”. I also do not see anything awkward related to conformation, it is just specific to steroids.

Answer: conclusion (iv) was removed

  1. Figure 7 presents conformation and directly copes Fig. 3 – Fig. 6. Instead, if similarity in conformation is the thing which is intended to be shown, make actual overlay of conformations so the identical parts are actually above one another. This can be done in, e.g., Discovery Studio and can be tried to achieved in Mercury as well.

Answer: The overlap of all four esters was added
e. Section 3.2. should be added as just one line to beginning of Section 3.1 indicating the agreement of XRPD (characterizing bulk sample) and from structure simulated XRPD (characterizing the analyzed crystal) showing that crystals represent the bulk sample. Figure 8 should be transferred to Supporting Information. The would deserve a place in the article only in case there would be complete mismatch between the patterns that would deserve some detailed explanation.

Answer:  Section 3.2 was moved to section 3.1 and Figures 8 were placed in Supporting information.
f. not directly related to crystallography but more to organic chemistry articles: Figures 1 and 2 would be better to join in one Structure presented in one figure by showing testosterone backbone with atom numbering and R in place of hydroxyl group attached to C17, and below the structure giving that R = OH in case of testosterone, COCH3 (could better be using structural formula) in case of acetate etc. Also, label chemical structures as “chemical structure” instead of “Structural/chemical perspective”.

Answer: Figures 1 and 2 were merged.
g. If the article would be published in crystallography journal, explanations regarding Hirshfeld surface could be reduced (e.g, reason why there are 2 surfaces for TBas, L324-327).

Answer: If possible, we would prefer it to be published in Materials Journal. If it is transferred to a crystallography journal, then we respect the given indications.
h. Figure 10 could be transferred to Supporting Information, as it is not discussed in the article text, but more easily interpretable 2D Fingerprint plot is the one being actually analyzed. Also the Table 3 showing information on mostly weak hydrogen bonds is not necessary in the text and I recommend to transfer it to Supporting Information.

Answer: Fig. 10 and Table 3 have been transferred to Supporting Information.

Minor comments:

  1. Introduction related to physiological action seems too long for such a crystallography paper.

Answer: Introduction related to physiological action was reduced.

  1. L51-53 in P2 does not seem necessary in a scientific article.

Answer: Lines 51-53 have been removed.

  1. “USP grade oils (they meet the requirements of United States Pharmacopeia)” repeats the same information twice, use either only “USP grade olis” or “Oils meeting the requirements of United States Pharmacopeia”

Answer: Only the indicated expression was used: “Oils meeting the requirements of United States Pharmacopeia”

  1. LynxEye is a position sensitive detector (L110)

Answer: This was completed

  1. As obtained from three measurements, add error bars (standard deviation or uncertainty) to Figure 11. I also strong;y encourage authors not to repeat information in Table 5 and Figure 11 and to transfer the former one to the Supporting Information.

Answer: Error bars have been added and Table 5 moved to Supporting Information.

  1. Provided it was one of the aims, give the absolute conformation for TPro and Tiso determined. Based on the conclusion “The Flack parameter shows that the propionate and acetate esters are pure enantiomers.” It seems that it possible that the third compound could not be pure enantiomer. Is it likely so or is it more plausible that this is just a result of some problems with crystal structure determination regarding this particular aspect?

Answer: The determination by anomalous dispersion is effective in compounds which contain lighter and heavier atoms as well. The negative value has no meaning and most likely appears because the errors.  Since it is difficult to determine the absolute configuration for compounds containing light elements (C, O, N) it is likely that the third compound is related to some problems regarding this particular aspect.

  1. As the structure of Testosterone acetate was not determined in this work, I suggest that Figure 3 can be put in Supporting Information. The most important structure data and its analysis can still be presented in the paper.

Answer: Figure 3 was moved to Supporting information

  1. There is a typo in Table 2 for value 33.3 of testosterone.

Answer: this was corrected

  1. In L384 if you write “On the other hand” you should have “On the one hand” somewhere before

Answer: The paragraph was corrected

Round 2

Reviewer 1 Report

The revised version of manuscript can be accepted for publication.

Author Response

Thank you again for your time and valuable suggestions that have led to the improvement of the manuscript.

Reviewer 2 Report

After the revision the manuscript seems to be a decent crystallography paper reporting crystal structures and their analysis for similar biologically relevant molecules. I still, however, don't see any justification for the choice of this journal.

In my view a couple of small changes still has to be made;

1. Comment on L201-203 about preferred orientation are not required, as they are self-evident.

2. L207-210. I believe that the value of Flack parameter confirm the correctness of the absolute configuration but not whether the crystal contain pure enantiomer. If a chiral molecule crystallizes in a chiral space group, this will be pure enantiomer, and the Flack parameter would just show how reliable is the determined absolute configuration. Therefore the conclusion "both solid forms are pure enantiomers" is wrong. The same applies also to the Conclusions L464-465.

Additional remark - the paper being published in non-crystallograhhy journal seems to have required adding explanation Coulomb-London-Pauli (CLP) approach, which is often used and would not require so long explanation in the crystallography field.

Author Response

Thank you once again for your time and for the valuable suggestions that we hope will lead to the improvement of the manuscript.

After the revision the manuscript seems to be a decent crystallography paper reporting crystal structures and their analysis for similar biologically relevant molecules. I still, however, don't see any justification for the choice of this journal.

Answer: We chose Materials because we have a discount in the processing of the manuscript.

In my view a couple of small changes still has to be made;

  1. Comment on L201-203 about preferred orientation are not required, as they are self-evident.

Answer: The paragraph was removed.

  1. L207-210. I believe that the value of Flack parameter confirm the correctness of the absolute configuration but not whether the crystal contain pure enantiomer. If a chiral molecule crystallizes in a chiral space group, this will be pure enantiomer, and the Flack parameter would just show how reliable is the determined absolute configuration. Therefore the conclusion "both solid forms are pure enantiomers" is wrong. The same applies also to the Conclusions L464-465.

Answer: The paragraphs have been corrected according to your instructions.

Additional remark - the paper being published in non-crystallograhhy journal seems to have required adding explanation Coulomb-London-Pauli (CLP) approach, which is often used and would not require so long explanation in the crystallography field.

Answer: Details of the computational methods have been included because Reviewer no 1 requested this.